# STOCHASTIC GRADIENT DISCRETE LANGEVIN DYNAMICS

## ABSTRACT

Sampling via Markov chain Monte Carlo can be inefficient when each evaluation of the energy function gradient depends on a large dataset. In continuous spaces, this challenge has been addressed by extending Langevin samplers with stochastic gradient estimators. However, such an approach cannot be directly applied to discrete spaces, as a naive application leads to biased estimates with large variance. To close this gap, we propose a new sampling strategy, *Stochastic Gradient Discrete Langevin Dynamics*, to provide the first practical method for stochastic distribution sampling in discrete spaces. The proposed approach mitigates the bias of naive "gradient" estimators via a novel caching scheme, and reduces estimation variance by introducing a modified Polyak step size scheme to adapt the simulation time. We demonstrate significant efficiency improvements across various sampling problems in discrete spaces, including Bayesian learning, stochastic integer programming, and prompt tuning for text-image models.

## 1 INTRODUCTION

Markov Chain Monte Carlo (MCMC) is one of the most widely used techniques to sample from complex and intractable probability distributions (Robert & Casella, 2013). In continuous spaces, gradient-based MCMC approaches such as Langevin Monte Carlo (Rossky et al., 1978), Hamiltonian Monte Carlo (Duane et al., 1987) and variants (Girolami & Calderhead, 2011; Hoffman et al., 2014) are able to accurately simulate the Langevin Dynamics (LD) and significantly improve sampling efficiency in both theory and practice (Cheng & Bartlett, 2018; Chen & Vempala, 2019; Carpenter et al., 2017). In discrete spaces, the Discrete Langevin Dyanmics (DLD) (Sun et al., 2023) has been recently proposed as a viable generalization of LD to discrete space dynamics, which leverages a characterization in terms of a continuous time Markov chain (ctMc). The "gradient-based" MCMC methods for discrete spaces, including LB (Zanella, 2020), GWG (Grathwohl et al., 2021), PAS (Sun et al., 2021; 2022), DLP (Zhang et al., 2022), and DLMC (Sun et al., 2023), can generally be interpreted as simulating the DLD.

A limitation of "gradient-based" MCMC methods is that they rely on computing the energy gradient, which can be extremely expensive when the energy is expressed in terms of an auxiliary expectation, such as an average over a massive dataset. Using stochastic approximation (Robbins & Monro, 1951), SGLD (Welling & Teh, 2011) and SGHMC (Chen et al., 2014) leverage noisy estimates of the energy gradient from mini-batches of the data to approximately simulate the Langevin dynamics. However, these methods are built upon a diffusion process and cannot be directly applied to discrete spaces for two major reasons: First, the accumulated transition kernel from a naive stochastic approximation does not converge to the DLD, even with infinitesimal simulation time. This is unlike the diffusion process, as the bias comes from the fact that the ratio estimator is not exchangeable with the expectation. Second, it is common to maintain a constant step size once it drops below a threshold (Welling & Teh, 2011; Chen et al., 2014), but in discrete spaces it is important to decrease the step size to zero, as the error of the stochastic approximation in DLD is typically many orders of magnitude larger than that in LD.

In this paper, we propose a new algorithm, *Stochastic Gradient Discrete Langevin Dynamics* (SGDLD), for efficiently simulating the DLD in discrete spaces when the energy differences are defined by an auxiliary expectation. Given that we focus on discrete spaces throughout this paper, we will abuse the terminology "gradient" to refer to the vector of local probability ratios (explained

in more detail below). SGDLD consists of two key techniques, gradient caching and step size adaptation, which each addresses one of the two issues above. First, we maintain a cache of the stochastic gradient approximations. When a step of DLD does not jump to a new state, the cached values are reused to correct the empirical ratio in the next step. In this way, we can effectively reduce the approximation error in the rate matrix with negligible computational overhead. Second, as the magnitude of the empirical probability ratios from different mini-batches can differ by several orders of magnitude, we introduce a modified Polyak step size scheme (Hazan & Kakade, 2019) in discrete spaces to automatically adapt the step size to accommodate different states and mini-batches. With proper annealing, we can prove that SGDLD samples from the correct distribution.

The highlights of the paper are organized as follows:

- In section 3, we define the sampling problem of stochastic distribution and point out its challenges.
- In section 4, we introduce stochastic gradient with caching and the Polyak step size to address the difficulties identified above, obtaining the SGDLD algorithm.
- In section 6, we empirically verify the theory in two synthetic sampling problems, and also apply SGDLD to three significant applications in stochastic integer programming, approximate computing, and prompt tuning for text to image models.

## 2 PRELIMINARIES

Our approach is built upon the foundations of Langevin Dynamics with extensions to stochastic gradient and discrete approximation.

**Langevin Dynamics**. The Langevin Dynamics (LD) describe a diffusion process where a point $x_t$ moves according to gradient ascent steps in $f(x)$ with Gaussian noise injected in the updates:

$$dX_t = \nabla f(X_t)dt + \sqrt{2}dW_t \tag{1}$$

such that $W_t$ is a Wiener process. The stationary distribution of the process in Equation 1 is $\pi(x) \propto \exp(f(x))$, and fast convergence of the process to the stationary distribution has been proved under various metrics (Durmus & Moulines, 2017; 2019; Cheng & Bartlett, 2018). Research on discrete time simulation of LD has delivered many efficient algorithms for sampling from a target distribution $\pi(x)$, such as LMC (Rossky et al., 1978) and the Unadjusted Langevin Algorithm (Parisi, 1981).

**Stochastic Gradient Langevin Dynamics**. Welling & Teh (2011) considered the important scenario of Bayesian learning, where $f(x)$ depends on a massive dataset $\mathcal{D}$. To avoid the huge computational cost in evaluating $\nabla f(x)$, Welling & Teh (2011) proposed SGLD, where the gradient $\nabla f(x)$ in Equation 1 is replaced by an unbiased stochastic approximation $\nabla \hat{f}(x)$. Given a time interval $h = N\epsilon$, SGLD simulates $N$ steps of the noisy LD, such that

$$x_{t+h} = x_t + \epsilon \sum_{n=1}^{N} \nabla \hat{f}(x_{t+n\epsilon}) + \sqrt{2}W_h \tag{2}$$

Denote $\hat{g}(x) = \hat{f}(x) - f(x)$. Under mild conditions that, for example, $\nabla \hat{f}(x)$ is $L$-Lipschitz and $\hat{g}(x)$ has finite variance $\sigma^2$, one can establish:

$$\epsilon \sum_{n=1}^{N} \nabla \hat{f}(x_{t+n\epsilon}) = \epsilon \sum_{n=1}^{N} \nabla \hat{f}(x_t) + O(h^2 L) = h\nabla f(x_t) + \epsilon \sum_{i=1}^{N} \nabla \hat{g}(x_t) + O(h^2 L) \tag{3}$$

When $N$ is sufficiently large, it is easy to verify that the second term of Equation 3 converges to a normal random variable with zero mean and variance $O(h\epsilon)$, which is dominated by $W_h$ with variance $O(h)$. Hence, Equation 2 reduces to:

$$x_{t+h} = x_t + h\nabla f(x_t) + \sqrt{2}W_h, \tag{4}$$

for sufficiently small $h$, showing that SGLD provides an asymptotically unbiased discrete time simulation of LD.

**Discrete Langevin Dynamics**. LD in a continuous space $\mathbb{R}^d$ can be generalized to a discrete Langevin dynamics (DLD) in a discrete space $\mathcal{X}$ as a continuous time Markov chain (ctMc) in

$\mathcal{X}$ (Sun et al., 2023) that satisfies:

$$\frac{d}{dt}\rho^t = \rho^t R, \quad R(x,y) = g\left(\frac{\pi(y)}{\pi(x)}\right) 1_{\{y \in N(x)\}}(x,y), \tag{5}$$

where $\rho^t \in \mathbb{R}^{|\mathcal{X}|}$ is the probability distribution of $X_t$ at time $t$, and $R \in \mathbb{R}^{|\mathcal{X}| \times |\mathcal{X}|}$ is the rate matrix. Here, $\pi(x)$ is the target distribution, $N(x)$ is the neighborhood set of $x$, and $g(\cdot) : \mathbb{R}_+ \to \mathbb{R}_+$ is a locally balanced weight function satisfying $g(a) = ag(\frac{1}{a})$, for example $g(a) = \sqrt{a}$ or $g(a) = \frac{a}{a+1}$ (Zanella, 2020). The gradient approximation $\frac{\pi(y)}{\pi(x)} \approx \exp(\langle \nabla \log \pi(x), y - x \rangle)$ has produced good performance in many applications (Grathwohl et al., 2021), so we adopt this as an intuitive "gradient" for discrete spaces.

## 3 SAMPLING FROM A STOCHASTIC DISTRIBUTION

### 3.1 STOCHASTIC DISTRIBUTION

Let $\pi(\cdot) : \mathcal{X} \to \mathbb{R}_+$ be an unnormalized distribution on $\mathcal{X}$. We assume querying the function $\pi(x)$ is expensive, for example $\pi(x)$ is a function that depends on a massive dataset. We say $\pi(\cdot)$ is a stochastic distribution if there exists functions $\psi(x, \xi) : \mathcal{X} \to \mathbb{R}$ and $\phi : \mathbb{R} \to \mathbb{R}$ such that:

$$\pi(x) = \phi(\mathbb{E}_\xi[\psi(x|\xi)]) \tag{6}$$

where $\phi$ and $\psi$ can be efficiently evaluated, and $\xi$ is a random variable that can be sampled efficiently. We give two concrete examples to illustrate this definition.

**Example: Quenched Model**. Assume $\pi(\cdot)$ has the form: $\pi(x) = \int p(u)\pi(x|u)du$, where $p(u)$ and $\pi(x|u)$ are easy to sample from. To rewrite $\pi(\cdot)$ as in Equation 6, we let $\xi = \{u_1, ..., u_B\}$ be a uniformly sampled mini-batch from $p(u)$, $\phi(\cdot)$ be the identity function, and $\psi(x|\xi) = \hat{\pi}(x|u_{1:B}) = \frac{1}{B}\sum_{i=1}^{B} \pi(x|u_i)$.

**Example: Bayesian Learning**. Assume $\pi(\cdot)$ has the form $\pi(x) = \exp\left(p(x) + \sum_{u_i \in \mathcal{D}} f(u|x)\right)$, where $\mathcal{D}$ is a dataset, $p(x)$ is a prior, and $f(u|x)$ is the likelihood function. To rewrite $\pi(\cdot)$ as in Equation 6, we let $\xi = \{u_1, ..., u_B\}$ be a uniformly sampled mini-batch from $\mathcal{D}$, $\phi(x) = \exp(x)$ and $\psi(x|\xi) = p(x) + \frac{D}{B}\sum_{i=1}^{n} f(u_i|x)$.

### 3.2 NAIVE STOCHASTIC GRADIENT

Let us assume that we have an unbiased noisy estimate of the rate matrix $\hat{R}$ for the DLD Equation 5, such that $\mathbb{E}[\hat{R}] = R$ and $\|\hat{R} - R\|_2 < U$ almost surely, where $U$ is a fixed constant. To simulate from $x_t$ to $x_{t+h}$, we split $h$ into $N$ smaller steps of duration $\frac{h}{N}$ and denote the stochastic rate matrix in each step $j$ as $R_j$. Since computing $\exp(\frac{h}{N}R_j)$ exactly is intractable, we follow Sun et al. (2023) to use

$$\exp(\frac{h}{N}R_j) \approx I + \frac{h}{N}R_j, \tag{7}$$

as the transition matrix in simulation. We only need to show that

$$\lim_{N \to \infty} \exp(hR) - \prod_{i=1}^{N}(I + \frac{h}{N}R_i) = 0 \tag{8}$$

Note that one can decompose $N = N_1 N_2$ to obtain:

$$\prod_{i=1}^{N}(I + \frac{h}{N}R_i) = \prod_{i=1}^{N_1}\prod_{j=1}^{N_2}(I + \frac{h}{N}R_{iN_1+j}) = \prod_{i=1}^{N_1}[I + \frac{h}{N_1}(\frac{1}{N_2}\sum_{j=1}^{N_2} R_{iN_1+j}) + O(\frac{1}{N_1^2})] \tag{9}$$

$$= \prod_{i=1}^{N_1}[\exp(\frac{h}{N_1}R) + O(\frac{1}{N_1\sqrt{N_2}}) + O(\frac{1}{N_1^2})] = \exp(hR) + O(\frac{1}{\sqrt{N_2}}) + O(\frac{1}{N_1}), \tag{10}$$

where $N_2$ controls the Monte Carlo estimation error from $\hat{R}$ to $R$, and $N_1$ controls the Taylor approximation error from $I + \epsilon R \approx \exp(\epsilon R)$. When both $N_1$ and $N_2$ are sufficiently large, the naive stochastic gradient DLD asymptotically converges to the correct target distribution $\pi(\cdot)$.

### 3.3 CHALLENGE IN DISCRETE SPACES

**Bias**: The derivation above assumes that the rate matrix $R$ has an unbiased estimator $\hat{R}$, which is not necessarily available for general discrete distributions. That is, in general:

$$g\left(\frac{\pi(y)}{\pi(x)}\right) = g\left(\frac{\phi(\mathbb{E}_\xi[\psi(x,\xi)])}{\phi(\mathbb{E}_\xi[\psi(x,\xi)])}\right) \neq \mathbb{E}_\xi\left[g\left(\frac{\phi(\psi(x,\xi))}{\phi(\psi(x,\xi))}\right)\right]. \tag{11}$$

This gap can also be illustrated in the two examples above. For simplicity, consider the weight function $g(a) = \sqrt{a}$.

In the Quenched Model scenario, the expectation clearly cannot be exchanged with the ratio

$$\sqrt{\frac{\pi(y)}{\pi(x)}} = \sqrt{\frac{\mathbb{E}_{u_{1:N}}[\hat{\pi}(y|u_{1:N})]}{\mathbb{E}_{u_{1:N}}[\hat{\pi}(x|u_{1:N})]}} \neq \mathbb{E}_{u_{1:N}}\left[\sqrt{\frac{\hat{\pi}(y|u_{1:N})}{\hat{\pi}(x|u_{1:N})}}\right].$$

In Bayesian Learning, Jensen's inequality reveals that the expectation is not generally exchangeable with the exponential:

$$\sqrt{\frac{\pi(y)}{\pi(x)}} = \exp\left(\mathbb{E}\left[\frac{M}{2B}\sum_{i=1}^{B} f(u_i|y) - f(u_i|x)\right]\right) \leq \mathbb{E}\left[\exp\left(\frac{M}{2B}\sum_{i=1}^{B} f(u_i|y) - f(u_i|x)\right)\right]$$

**Magnitude**: Besides the bias in $\hat{R}$, its large variance requires an extremely small simulation time $\epsilon$, which slows the mixing rate of the algorithm with increasing iterations. This is unlike the continuous case, where Welling & Teh (2011) and Chen et al. (2014) keep the step size constant once it has decreased below a threshold, since when the threshold is sufficiently small the MH rejection rate is negligible. Unfortunately, in the discrete case, choosing a fixed threshold for all states and all mini-batches is typically not a good choice. The first order Taylor approximation of $\exp(\epsilon\hat{R})$ in Equation 7 requires the simulation time $\epsilon$ in the order of $O(\|\hat{R}\|^{-1})$. However, given that $\hat{R}$ consists of probability ratios, Equation 5, that are an exponential of the gradients $\nabla f(x)$ used in LD, the resulting norms of the stochastic rate matrix $\hat{R}$ across mini-batches in the discrete case can differ by orders of magnitude.

For example, we evaluated the probability ratio in the Bayesian logistic regression task below (see Section 6.2) with 200 different mini-batches. The magnitude of the gradient and the jump rate are visualized in Figure 1 (see $Z(x)$ in Equation 14 for the definition of the jump rate). The gradient norms are all of the same order on the log scale while the largest jump rate can be $10^{30}$ times of the smallest jump rate. Choosing a simulation time $\epsilon$ that is small enough for the largest jump rate will cause the process to be stuck for an inordinately long time at locations with a small jump rate.

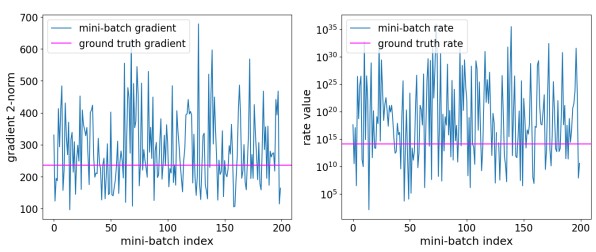

Figure 1: Scale for Gradients and Jump Rates

**Algorithm 1** One Step of SGDLD

1: **Input**: state $x_t$, cache $\mathcal{C}$
2: $\mathcal{C} \leftarrow \mathcal{C} \cup \{\psi(z,\xi) : z \in N(x_t)\}$
3: Get rate $\hat{R}$ via Equation 12.
4: Get simulation time $\epsilon_t$ via Equation 14.
5: Sample new state $x_{t+1} \sim I + \epsilon_t\hat{R}$.

6: **if** $x_{t+1} \neq x_t$ **then**
7:    Empty cache $\mathcal{C}$.
8: **end if**
9: **Return**: $x_{t+1}, \epsilon_t$

## 4 STOCHASTIC GRADIENT DISCRETE LANGEVIN DYNAMICS

### 4.1 STOCHASTIC GRADIENT WITH CACHE

The most straightforward approach to address the bias in $\hat{R}$ is to increase the batch size $B$ but this also increases the computational cost. Moreover, when a large batch size mitigates the estimation

error Equation 10 controlled by $N_1$, the error from first order Taylor approximation of $\exp(\epsilon \hat{R})$ still requires a large $N_2$. As a result, the algorithm still only admits a small simulation time $\epsilon$, where the new state remains in the same position $x_{t+\epsilon} = x_t$ with a high probability under the transition matrix $I + \epsilon \hat{R}$. In this case, the information collected at $x_t$ is lost. This is unlike SGLD (Welling & Teh, 2011) in a continuous space, where an update of the state $x_{t+\epsilon} \neq x_t$ occurs in every step.

To address this inefficiency, conveniently, whenever the DLD stays at the same state during a simulation interval $\epsilon$, i.e., $x_{t+\epsilon} = x_t$, we can cache the information collected at $x_t$ and reuse it to compute the rate matrix at $x_{t+\epsilon}$. Assume that a sequence of states $x_0 = x_\epsilon = ... = x_{(m-1)\epsilon} = x$ has occurred in the DLD without a jump. We can maintain the cache $\mathcal{C} = \{\psi(z, \xi_k) : z = x \text{ or } z \in N(x)\}_{k=0}^{m-1}$, where $\xi_k = u_{kB+1:(k+1)B}$ is the mini-batch collected at state $x_{k\epsilon}$. The empirical probability ratio $\frac{\hat{\pi}_k(y)}{\hat{\pi}_k(x)}$ from $x_{k\epsilon}$ to $x_{(k+1)\epsilon}$ can then be calculated as:

$$\frac{\hat{\pi}_k(y)}{\hat{\pi}_k(x)} = \frac{\phi(\frac{1}{m}\sum_{i=1}^{m}\psi(y, \xi_i))}{\phi(\frac{1}{m}\sum_{i=1}^{m}\psi(x, \xi_i))}. \tag{12}$$

Again, we use the two concrete examples above to illustrate the empirical probability ratio:

**Quenched:** $\frac{\hat{\pi}_k(y)}{\hat{\pi}_k(x)} = \frac{\sum_{i=1}^{mB}\pi(y|u_i)}{\sum_{i=1}^{mB}\pi(x|u_i)}$, **Bayesian:** $\frac{\hat{\pi}_k(y)}{\hat{\pi}_k(x)} = \exp\left(\frac{M}{mB}\sum_{i=1}^{mB}f(u_i|y) - f(u_i|x)\right).$

The caching technique can effectively expand the batch size without increasing the computation. With some mild assumptions, Proposition 4.1 shows that the stochastic gradient sampler indicated by Equation 12 is asymptotically unbiased.

**Proposition 4.1.** *Assume for all $x, y, u$, the likelihood ratio $\frac{\pi(y;u)}{\pi(x;u)}$ is bounded by a fixed value $U$. Then, when the step size $\epsilon$ decreases to 0, the sampling process associated with jump rate from Equation 12 is asymptotically unbiased.*

See the complete proof in Appendix A.

**Remark**: Similar to Hamiltonian Monte Carlo, the sampling process above has an equivalent form of memoryless Markov chain. See more detailed discussion in Appendix B.1.

### 4.2 POLYAK STEP SIZE

Although we have an asymptotically unbiased estimator, as mentioned in Section 3.3, the large deviation of the magnitude of $\hat{R}$ makes the stochastic simulation of DLD very unstable. To address this problem, we borrow an idea from Polyak step adaptation in convex optimization (Hazan & Kakade, 2019). Given an objective function $f(x)$, a Polyak step in a gradient descent with step size $\eta_t$ is given by:

$$x_{t+1} = x_t - \eta_t \nabla f(x_t), \quad \eta_t = \frac{h_t}{\|\nabla f(x_t)\|^2}, \tag{13}$$

where $h_t$ is hand designed schedule and the norm of the gradient is used to normalize the step size. We extend this idea to discrete spaces, where we can also maintain a Polyak step $\epsilon_t$. Given a designed schedule $h_t$ and current state $x_t$, let:

$$\epsilon_t = \frac{h_t}{Z(x_t)}, \quad Z(x) = \sum_{z \in N(x)} g\left(\frac{\pi(z)}{\pi(x)}\right), \tag{14}$$

where the value $Z(x)$ is the *jump rate* for leaving the current state $x$. We use $Z(x)$ in place of the norm of gradients for simulating DLD. In this way, the step size $\epsilon_t$ can be automatically adjusted for all states $x$ and all mini-batches. In practice, calculating the probability ratio $\frac{\pi(z)}{\pi(x)}$ in the *jump rate* $Z(x)$ exactly could be time consuming. Hence, we follow Grathwohl et al. (2021) to use a gradient approximation. Empirically, we find this is sufficient to stabilize the sampling process.

For the Polyak step size schedule, we can set a threshold $h^*$ and gradually decrease $h_t$ until it reaches $h^*$. In this way, the Monte Carlo estimation for a function $F(x)$ of interest is

$$\mathbb{E}_x[F(x)] = \frac{\sum_{t=1}^{T}\epsilon_t F(x_t)}{\sum_{t=1}^{T}\epsilon_t}. \tag{15}$$

We name this algorithm as *Stochastic Gradient Discrete Langevin Dynamics* (SGDLD) and summarize it in Algorithm 1.

## 5 RELATED WORK

In scaling to massive or streaming datasets, which has become more prevalent in recent years, stochastic gradient variants of dynamics-based samplers have achieved significant success. Stochastic gradient Langevin dynamics (SGLD) (Welling & Teh, 2011) attempted the first step in this direction. Following this work, Ahn et al. (2012) proposed to use Fisher scoring as a pre-conditioning matrix in SGLD to accelerate mixing. Patterson & Teh (2013) generalized SGLD to the probability simplex by choosing a proper Riemannian metric. Chen et al. (2014) proposed the stochastic gradient Hamiltonian Monte Carlo (SGHMC) algorithm and introduced a friction term to address the entropy explosion problem. Finally, Baker et al. (2019) incorporated a variance reduced stochastic gradient estimator in SGLD for faster sampling. These methods are fundamentally based on diffusion processes and suitable for continuous spaces.

However, for discrete spaces, the corresponding theories are far less well understood. Recently, Zanella (2020) introduced an informed proposal for discrete spaces, and proved that a family of locally balanced functions is asymptotically optimal. Inspired by this work, Sun et al. (2023) generalized the Langevin dynamics to discrete spaces by introducing DLD as a continuous time Markov chain, showing that the various locally balanced proposal samplers (Grathwohl et al., 2021; Sun et al., 2021; Zhang et al., 2022; Sun et al., 2022) are a discretizations of the DLD. Mimicking SGLD, Zhang et al. (2022) attempted to generalize SGLD under the assumption that an unbiased estimator of the rate matrix exists. However, the continuous time Markov chain perspective followed in this work is fundamentally different from the diffusion process, hence a corresponding understanding of the discrete Langevin dynamics for stochastic distributions, as considered in this paper, was lacking.

## 6 MORE EXPERIMENT DETAILS

In this section, we evaluate *stochastic gradient discrete Langevin Dynamics* (SGDLD) on two synthetic tasks and three real-world applications. For an abalation study, we also consider two variants: (1) SGDLD-noC, which does not use the gradient caching technique Equation 12, and (2) SGDLD-noP, which does not use the Polyak step size Equation 14. We omitted the results for SGDLD-noP in three applications as it can not generate reasonable solutions.

### 6.1 GAUSSIAN BERNOULLI MODEL

We first validate SGDLD on a simple quenched model with $x \in \{0, 1\}^4$. We let the auxiliary variable $u \in \mathbb{R}^4$ satisfy a Gaussian mixture model

$$u \sim \sum_{i=1}^{16} w_i \mathcal{N}(\mu_i, \Sigma_i), \tag{16}$$

and let the likelihood $\pi(x|u) \propto \exp(\langle x, u \rangle)$ satisfy a Bernoulli model. In this case, we can exactly compute the probability distribution for $x$. See more details in Appendix C.1.

We measure the distance between the empirical distribution obtained from samples and the true probability distribution to quantify sample quality. We compare SGDLD with SGDLD-noC and Gibbs, which computes the empirical conditional probability based on the mini-batch in each step. For all methods, we use ⟨method⟩-$b$ to denote the batch size $b$ that is used. We report the mean and standard deviation of the total variation in Figure 2. The results strongly support the claims made in this paper.

- SGDLD algorithm has smaller total variation than Gibbs.
- The total variation of SGDLD is consistently less than SGDLD-noC with the same batch size, implying the gradient cache can reduce bias.
- The total variation of SGDLD decreases when the step size decreases, which is consistent with the claim that SGDLD is asymptotically unbiased.
- The total variation of SGDLD-noC does not decreases when the step size decreases, which implies SGDLD-noC is not asymptotically unbiased.

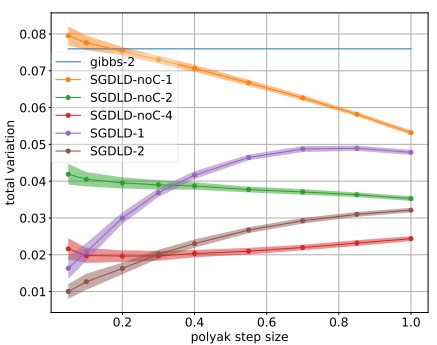

Figure 2: Total Variation Distance on Gaussian Bernoulli Model.

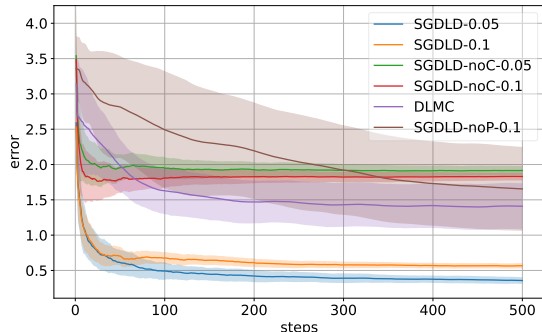

Figure 3: Estimation Error of variable selection using Bayesian Logistic Regression.

Another interesting phenomenon is that the total variation of SGDLD-noC-1 and SGDLD-noC-2 increases when we decrease the step size. The variation is dominated by the Monte Carlo estimation error when Polyak step size is small, and is dominated by the approximation error Equation 7 when Polyak step size is large. Results with larger batch size are given in Appendix C.1.

## 6.2 BAYESIAN LOGISTIC REGRESSION

We apply SGDLD to Bayesian logistic regression for variable selection. Specifically, for a dataset $X \in \mathbb{R}^{m \times d}$ and $Y \in \{0, 1\}^m$ we assume the log likelihood function is given by:

$$f(Y|X) = \langle Y, X\beta \rangle - \log \text{Sigmoid}(X\beta) \tag{17}$$

where the binary vector $\beta \in \{0, 1\}^d$ plays the role of selecting variables. Details in Appendix C.2.

For baselines, we consider SGDLD-noC, SGDLD-noP, and DLMC (Sun et al., 2023) that uses the entire dataset in every MH step. For the stochastic methods, we denote each as $\langle \text{method} \rangle$-$h$, where $h$ is the threshold for the Polyak step size. For SGDLD-noP-$h$, the simulation time threshold is selected so that is has the same average jump distance as SGDLD-$h$. For DLMC, we use the optimal simulation time obtained by tuning the average acceptance rate to match 0.574 (Sun et al., 2022).

To measure the mixing rate, we report the 1-norm estimation error of the marginal distribution of $\beta$. To calibrate the computation cost, each step in the x-axis of Figure 3 refers to 320 updates for the stochastic methods and 2 updates for DLMC. The results in Figure 3 strongly support our claims:

- SGDLD has a faster mixing rate than DLMC, as SGDLD requires less computation in each step.
- SGDLD has significantly smaller estimation error than SGDLD-noC. The reason is SGDLD leverages gradient caching to correct the transition probability.
- SGDLD has a faster mixing rate than SGDLD-noP. Also, SGDLD has smaller variance as it is less affected by the large variance in the stochastic approximation of the jump rate.

We conducted extra experiments to show the fast mixing of SGDLD and demonstrate its advantage compared to more baselines, such as pseudo marginal MCMC (Bardenet et al., 2017). The results are given in Appendix C.2.

## 6.3 STOCHASTIC FACILITY LOCATION

Stochastic mixed integer programming (SMIP) — which integrates two hard optimization problems, integer programming (Conforti et al., 2014) and stochastic programming (Prékopa, 2013) — poses problems that are typically hard to solve (Sen, 2005). A commonly used approach is to combine sample average approximation (SAA) (Kleywegt et al., 2002) with Bender's decomposition (BnnoBRs, 1962), which is restricted to a finite number of samples due to the hardness of integer programming. Here, SGDLD can find a high quality solution by efficiently evaluating a massive number of samples.

Table 1: Facility Location with Stochastic Demands

| Problem Size | $15 \times 30$ | | $40 \times 120$ | | $100 \times 400$ | |
|---|---|---|---|---|---|---|
| Method | Cost $\downarrow$ | Time (s) | Cost $\downarrow$ | Time (s) | Cost $\downarrow$ | Time (s) |
| Gurobi (8) | $10418 \pm 499$ | 2 | $29390 \pm 3018$ | 256 | $73342 \pm 3436$ | 3502 |
| Gurobi (1024) | $10152 \pm 494$ | 103 | $26137 \pm 2428$ | 3342 | $68924 \pm 4727$ | 3621 |
| SLS | $9981 \pm 480$ | 16 | $26514 \pm 2283$ | 25 | $73849 \pm 3036$ | 91 |
| SGDLD-noC | $9870 \pm 487$ | 15 | $25773 \pm 2576$ | 25 | $67532 \pm 2983$ | 96 |
| SGDLD (Ours) | $\mathbf{9792} \pm 515$ | 15 | $\mathbf{25398} \pm 2366$ | 26 | $\mathbf{66395} \pm 3181$ | 98 |

Table 2: Relative errors of different methods with the AxC unit constraint as 3,5,8

| Threshold | RL | GS-Tr+S | GS-Tr+R | CON | AFF | SGDLD-noC | SGDLD | OPT |
|---|---|---|---|---|---|---|---|---|
| 3 AxC units | 7.68 | 4.87 | 3.24 | 3.18 | 3.10 | 3.34 | **2.95** | 2.77 |
| 5 AxC units | 10.15 | 8.03 | 5.86 | **5.13** | 5.38 | 5.26 | **5.13** | 4.74 |
| 8 AxC units | 12.83 | 12.65 | 10.62 | 10.17 | 10.04 | 9.93 | **9.81** | 8.56 |

We consider the facility location problem with stochastic demand (Albareda-Sambola et al., 2011; Bieniek, 2015). Let $I$ and $J$ denote the index sets for facilities and customers, respectively. Denote $y_i \in \{0, 1\}$ as whether facility $i \in I$ is open or not, $x_{ij} \in \{0, 1\}$ as whether customer $j \in J$ is served by facility $i$, and $s_i$ as the outsource for facility $i$. The objective function is:

$$f(x, y; d) = \sum_{i \in I} c_i y_i + \sum_{i \in I, j \in J} c_{ij} d_j x_{ij} + \sum_{i \in I} g_i s_i, \tag{18}$$

More details about the parameters $c_i$, $c_{ij}$, $d_j$ and $g_i$ are given in Appendix C.3. We use $|I| \times |J|$ to refer to the size of the problems we consider. We can transform the optimization problem into a sampling problem by considering the following probability distribution

$$\pi(x, y; \tau) \propto \exp(-\beta \mathbb{E}_d[f(x, y; d)]), \tag{19}$$

where $\beta$ is the inverse temperature used to control the smoothness of $\pi(\cdot)$. In this case, sampling is equivalent to Bayesian learning with a dataset having infinite samples $d$. We compare SGDLD with SGDLD-noC, SGDLD-noP, Gurobi 10.0 (Bixby, 2007) and stochastic local search (SLS) (Hoos & Stützle, 2004). For Gurobi, we use SAA with 8 and 1024 samples. For SLS, we use the same procedure as SGDLD except replace the sampling step by greedily picking the best local edit in the neighborhood. We solve problems at three different sizes. After each method returns a configuration $(x, y)$, we sample another 10k demands $d$ and report the average cost with standard deviation in Table 1. The results show that SGDLD significantly outperforms other methods in all sizes.

### 6.4 APPROXIMATE COMPUTING

SGDLD can also be used to solve black-box stochastic integer optimization. For example, one fundamental problem in approximate computing (AxC) is to assign imprecise functional units (a.k.a. AxC units) to execute operations such as multiplication or addition (Han & Orshansky, 2013; Mittal, 2016), aiming to significantly reduce circuit energy with tolerable error. We follow Wang et al. (2022) and formulate the problem as a computation graph that has 15 nodes of multiplication or addition that maps $\mathbb{R}^{16}$ to $\mathbb{R}$. The random variable $w$ determines the computational task to execute. The energy function is defined as the expectation of the computing error $\mathbb{E}_w[f(x; w)]$. More details about the problem are given in Appendix C.4

Similar to the stochastic facility location problem, we can transform the optimization problem into a sampling problem by considering the energy based model:

$$\pi(x) \propto \exp(-\beta \mathbb{E}_w[f(x; w)]) \tag{20}$$

We report the average relative error for SGDLD in Table 2. For the other methods, we use the number reported in (Wang et al., 2022). We can see that SGDLD has comparable or better performance than state-of-the-art learning based methods CON and AFF. Computationally, sampling based method can be much cheaper. In particular, SGDLD only requires 10k evaluations of $f(x; w)$ to generate a solution. For CON and AFF, ignoring the training and inference cost, collecting training data alone requires more than 100 million evaluations (Wang et al., 2022).

Table 3: Prompt Tuning for Style Transfer. For each method and prompt length, we run the algorithm for 10 times and report the mean and standard deviation of the CLIP (Radford et al., 2021) similarity ($\uparrow$) between the best text prompts found and the target images.

| Prompt Length | 4 | 8 | 16 | 32 |
|---|---|---|---|---|
| SGDLD | $\mathbf{0.4201} \pm 0.0089$ | $\mathbf{0.4452} \pm 0.0071$ | $\mathbf{0.4708} \pm 0.0127$ | $\mathbf{0.4844} \pm 0.0095$ |
| SGDLD-noC | $0.3689 \pm 0.0166$ | $0.3655 \pm 0.0138$ | $0.3670 \pm 0.0141$ | $0.3678 \pm 0.0161$ |
| CR Wen et al. (2023) | $0.3957 \pm 0.0054$ | $0.4195 \pm 0.0063$ | $0.4375 \pm 0.0060$ | $0.4526 \pm 0.0049$ |

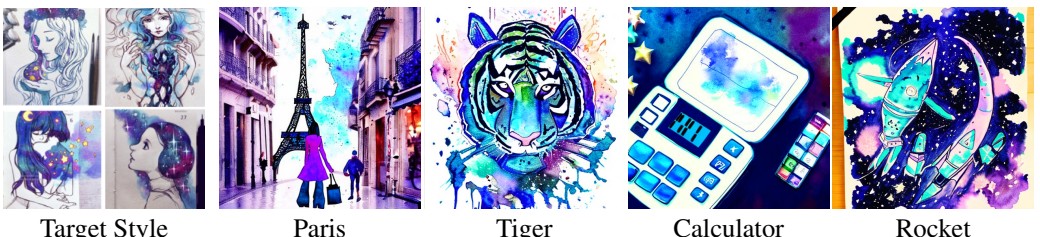

| Target Style | Paris | Tiger | Calculator | Rocket |

Figure 4: Examples of images generated using SGDLD with prompt length 32

## 6.5 PROMPT TUNING

The performance of large language models (Chowdhery et al., 2022; OpenAI, 2023) and diffusion generative models (Rombach et al., 2022; Radford et al., 2021) can heavily depend on the quality of the prompts (Lester et al., 2021; Wei et al., 2022). Since the text prompts are discrete, the SGDLD algorithm is a natural choice to sample good prompts. We follow the style transfer experiments in Wen et al. (2023). In particular, given a target style represented by a set of target images $\mathcal{I} = \{I_1, ..., I_n\}$, we sample text prompts $x$ that obtain a high CLIP (Radford et al., 2021) score on the target images. Similar to last two applications, we consider the target distribution:

$$\pi(x) = \exp(\beta \mathbb{E}_{I \sim \mathcal{I}}[\mathrm{CLIP}(x, I)]). \tag{21}$$

We compare SGDLD with the continuous relaxation (CR) method in Wen et al. (2023) and SGDLD-noC. The CLIP similarity is reported in Table 3 and some examples generated by SGDLD are given in Figure 4. In Table 3, one can see that SGDLD obtains a CLIP similarity that is consistently larger than the other two methods. More details and examples are given in Appendix C.5.

## 7 DISCUSSION

In this work, we generalize the stochastic gradient Langevin dynamics to discrete spaces. The proposed approach builds on the foundation of discrete Langevin dynamics, but uses stochastic approximations of the probability ratio to avoid calculating over an entire dataset. The proposed algorithm incorporates significant advances, since naive implementations will be biased in probability ratio estimation and also unstable in selecting the simulation time threshold. The proposed SGDLD addresses these two challenges by introducing novel gradient caching scheme and a Polyak step size control to deliver an asymptotically unbiased algorithm. Empirically, the method demonstrates good performance in both stochastic sampling tasks and stochastic optimization problems.

Despite the advances in SGDLD, there remains plenty of room to improve stochastic sampling in discrete spaces. In Equation 12, we implement a very simple cache approach. We believe that introducing variance reduction techniques will help improve sampling efficiency. Also, SGDLD currently simulates discrete Langevin dynamics in an unadjusted manner, requiring an exact calculation of the probability ratio to ensure asymptotic unbiasedness. For more challenging scenarios where one does not have access to the exact probability ratio and gradient approximation is required, unbiasedness is not guaranteed. Deriving an MH rejection step based on mini-batch data is a possible solution. Nevertheless, this work provides a first viable step toward developing efficient MCMC samplers for discrete spaces based on stochastic approximation, and we will seek further improvements in future work.

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

## A  DEFERRED PROOF

**Proposition A.1.** *Assume for all $x, y, u$, the likelihood ratio $\frac{\pi(y;u)}{\pi(x;u)}$ is bounded by a fixed value $U$. Then, when the step size $\epsilon$ decreases to 0, the sampling process associated with jump rate from Equation 12 is asymptotically unbiased.*

*Proof.* For a given time interval $h$, consider a split into $N$ steps so that $h = N\epsilon$. Denote the batch size as $B$. For each step $k = 0, ..., N - 1$, the samples collected at step $k$ can be indexed as $u_{kB+1:(k+1)B}$. For $0 \le i < j \le N - 1$, let $R_{i:j}$ denote the rate matrix computed using $u_{iB+1:jB}$. By assumption Equation 6, we have that $R_{i:j}$ is an asymptotically unbiased estimator of $R$ as $j - i$ increases to infinity. For simplicity, let $\hat{P}(x_s \to x_t)$ denote the empirical transition probability from time $s$ to time $t$.

To compute the accumulated transition probability for the first two steps, we need to consider two cases: First, with probability $1 - O(\epsilon)$, the state does not jump in the first step and the accumulated transition probability is $(I + \epsilon R_{0:1})(I + \epsilon R_{0:2})$. Second, with probability $O(\epsilon)$, the state jumps in the first step and the accumulated transition probability is $(I + \epsilon R_{0:1})(I + \epsilon R_{1:2})$. Combining these two cases yields the transition probability over the first two steps:

$$\hat{P}(x_0 \to x_{2\epsilon}) = I + \epsilon(R_{0:1} + R_{0:2}) + O(\epsilon^2) \tag{22}$$

Then, applying Equation 22 recursively to the remaining steps, we have accumulated transition probability after $N$ steps as:

$$\hat{P}(x_0 \to x_h) = I + h\frac{1}{N}\sum_{i=1}^{N} R_{0:i} + O(h\epsilon) \tag{23}$$

Since $R_{0:i}$ converges to $R$, we have:

$$\lim_{h \to 0}\lim_{N \to \infty} \frac{\hat{P}(x_0 \to x_h) - \exp(hR)}{h} = 0, \tag{24}$$

which implies that as $\epsilon$ decreases to 0, the empirical transition probability is asymptotically unbiased. $\square$

## B  DISCUSSION

### B.1  MEMORYLESS MARKOV CHAIN

Assume we have an initial state $X_1 = z_1$. Using the caching scheme, we generate a chain

$$(X_0 = z_0, \epsilon_0), (X_1 = z_0, \epsilon_1), \dots, (X_{n_1-1} = z_0, \epsilon_{n_1-1}), (X_{n_1} = z_1, \epsilon_{n_1}), (X_{n_1+1} = z_1, \epsilon_{n_1+1}), \dots$$

That is to say, the changes of the state occur at steps $0 = n_0 < n_1 < \dots$, while the intermediate steps made no change and thus cached. Superficially this looks like a non-Markovian chain, but we can condense this sequence and obtain a Markov chain:

$$(Y_0 = z_0, \varepsilon_0), (Y_1 = z_1, \varepsilon_1), \dots, \quad \text{where } \varepsilon_j = \sum_{i=n_{j-1}}^{n_j} \epsilon_i \tag{25}$$

Specifically, the sequence $X_{n_j}, X_{n_j+1}, \cdots, x_{n_{j+1}-1}$ are condensed into a single state $Y_j$. For any objective function $f(\cdot)$, the Monte Carlo estimation on the stationary distribution $\pi(\cdot)$ is:

$$\mathbb{E}_{z \sim \pi(\cdot)}\left[\frac{1}{\sum_{i=1}^{m} \varepsilon_i}\sum_{i=1}^{m} \varepsilon_i f(z_i)\right] \tag{26}$$

In another word, the process to generate the sequence $X_{n_j}, X_{n_j+1}, \cdots, X_{n_{j+1}-1}$ is simply a proposal and the transition between $(Y_j, \varepsilon_j)$ to $(Y_{j+1}, \varepsilon_{j+1})$ are Markovian.

In analogy, in the Hamiltonian Monte Carlo algorithm, one uses the leapfrog algorithm to simulate the Hamiltonian dynamics in multiple steps to give a proposal. The non-Markovian within the proposal does not harm the to get a memoryless Markov chain in the end.

## C  PROBLEM FORMULATION IN EXPERIMENT

The experiments are run on a GCP virtual machine, with n1-standard-32 Intel Haswell CPU, 8 Nvidia V100 GPUs. We run 100 chains in C.1, C.2, C.3, 1 chain in C.4, and 10 chains in C.5. More details see below.

For sampling tasks in the Gaussian Bernoulli Model and Bayesian Logistic Regression, we use step size $h_t = \text{clip}(3 \cdot (10 + t)^{-0.5}, \min = h_{\min})$ In sampling tasks, the results reported in Figures 2 and 3 only use samples collected with $t$ large enough, such that $h_t = h_{\min}$. Hence, the schedule does not affect the result. For optimization tasks, we use $h_t = \text{clip}(\frac{1000}{100+t}^{0.5}, \min = 0.3)$ in facility location, $h_t = \text{clip}(\frac{3}{10+t}^{0.5}, \min = 0.1)$ in approximate computing, and $h_t = 1$ in prompt tuning. In optimization tasks, we did some hyperparameter searching and we found that, if $h_t$ decreases too fast or too slow, SGDLD will need more steps to obtain good solutions.

### C.1  GAUSSIAN BERNOULLI MODEL

We generate the Gaussian Bernoulli Model via Gaussian integral trick (Hubbard, 1959). Specifically, we consider a Markov random field with distribution:

$$\pi(x) \propto \exp(\frac{x^T W x}{2} + b^T x) \tag{27}$$

where:

$$W = \begin{bmatrix} 1 & 0.5 & 0.5 & 0.5 \\ 0.5 & 1 & 0.5 & 0.5 \\ 0.5 & 0.5 & 1 & 0.5 \\ 0.5 & 0.5 & 0.5 & 1 \end{bmatrix}, \quad b = W \begin{bmatrix} -0.5 \\ -0.5 \\ -0.5 \\ -0.5 \end{bmatrix} \tag{28}$$

Then, we introduce auxiliary variable $u$ such that $\pi(u|x) = \mathcal{N}(W^{\frac{1}{2}}x, I)$. Then, we have the marginal distribution of $u$ is a Gaussian mixture model

$$u \sim \sum_x \pi(x)\mathcal{N}(W^{\frac{1}{2}}x, I) \tag{29}$$

and the condition distribution of $x$ is Bernoulli:

$$\pi(x|u) \propto (x^T(W^{\frac{1}{2}}u + b)) \tag{30}$$

Results with larger batch size are given in Figure 5. One can see that caching scheme consistently improve the sampling efficiency.

### C.2  BAYESIAN LOGISTIC REGRESSION

In Bayesian logistic regression, for dataset $X \in \mathbb{R}^{m \times d}, Y \in \{0, 1\}^m$, we assume the log likelihood function is given by:

$$f(Y|X) = \langle Y, X\beta \rangle - \text{logSigmoid}(X\beta) \tag{31}$$

and the binary vector $\beta \in \{0, 1\}^d$ plays the role of selecting variables. Following Zhou (2020), we sample $X_i \sim \mathcal{N}(0, (I + \mathbf{1}\mathbf{1}^T)/d)$ for $i = 1, ..., m$ independently. We set $\beta_j = 1$ if $j < \sqrt{d}$ and $\beta_j = 0$ for else. Then, we sample $Y_i \sim \text{Bernoulli}(\text{sigmoid}(X_i\beta))$ for $i = 1, ..., m$ independently. After creating the dataset $(X, Y)$, we expand the model dimension by duplication, i.e., $\tilde{X}_i = \text{cat}([X_i, X_i]) \in \mathbb{R}^{m \times 2d}$, to make the problem more challenging (Titsias & Yau, 2017). As a result, in the new selector $\tilde{\beta} \in \{0, 1\}^{2d}$, the last $d$ dimensions have the same probability to be set to 1 as the first $d$ dimensions. In main text, we consider $2d = 18$ and $m = 100k$.

We conducted extra experiments with $2d = 100$ and $m = 100k$ with more baselines, for example, pseudo-marginal MCMC (Bardenet et al., 2017). In continuous spaces, pseudo marginal samplers use an uninformed proposal distribution and various variance reduction techniques to control the MH acceptance rate. However, these variance reduction techniques do not directly apply to problems in discrete spaces. To enable comparison, we consider a random walk oracle, which is a Random Walk Metropolis that has access to the exact energy function while having the same computation cost as our SGDLD. Such a random walk oracle provides an upper bound of the efficiency for pseudo-marginal samplers in discrete space. The results are given in Figure 6. One can see that, though with

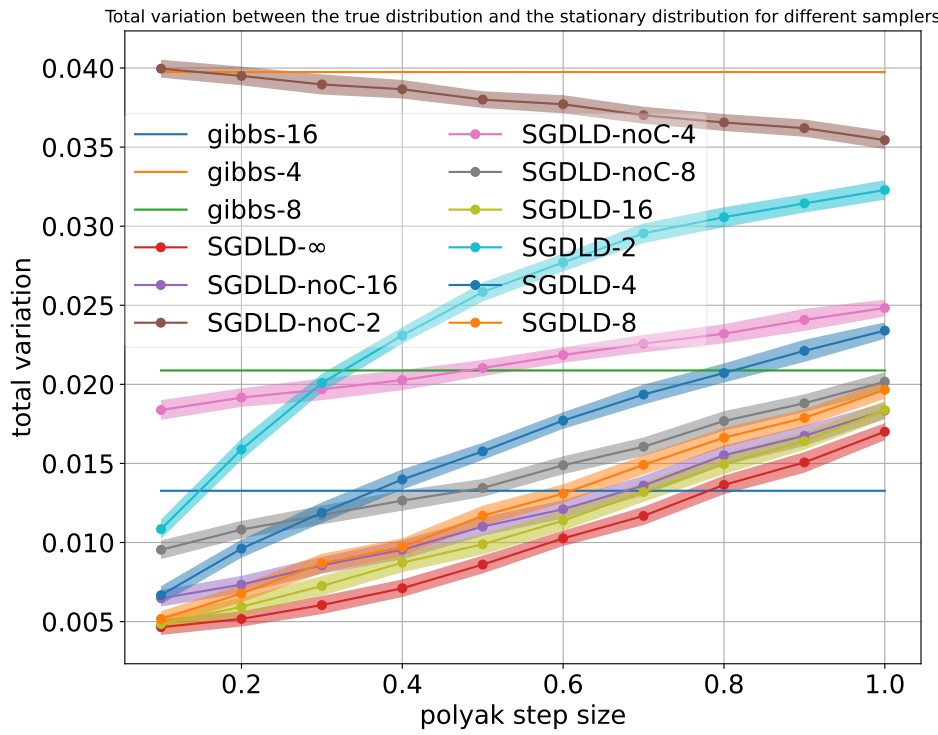

Figure 5: Total Variation on Gaussian Bernoulli Model

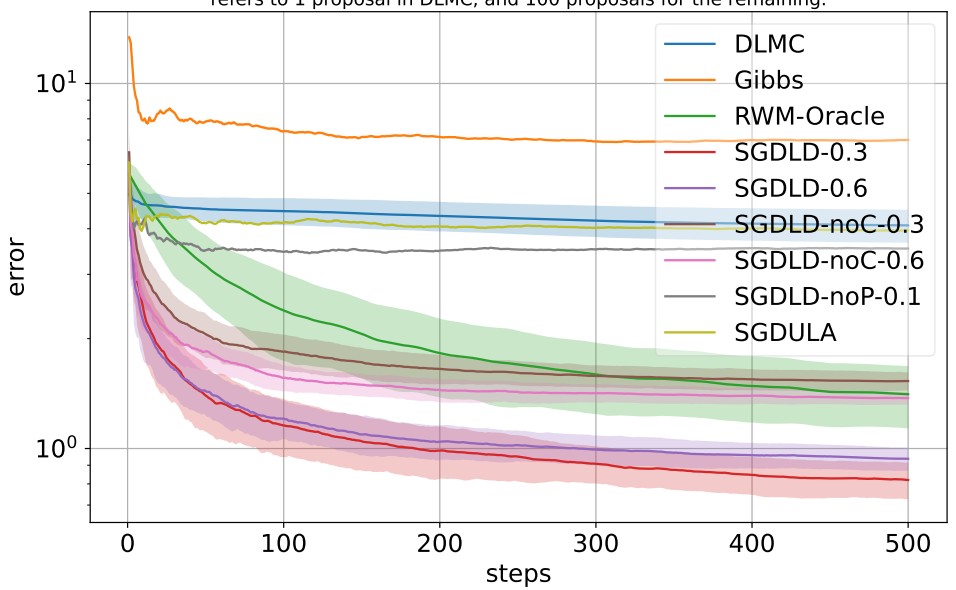

Figure 6: Extra Experiments on Logistic Regression

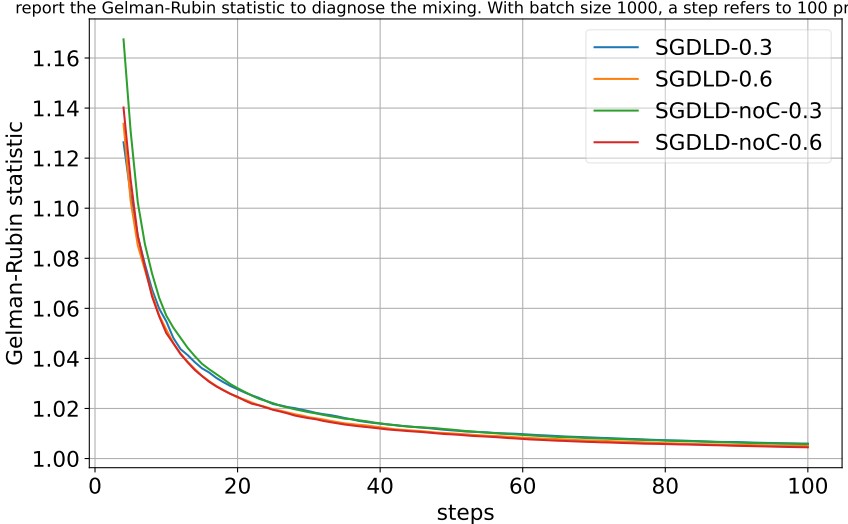

Figure 7: Diagnosis of Mixing

a zero variance estimator, the RWM-Oracle has a larger estimation error than SGDLD. The reason is that a pseudo-marginal sampler is uninformed, thus it mixes slower than an informed sampler.

We also examine the Gelman-Rubin statistic (Gelman & Rubin, 1992) as a diagnosis of mixing. When the statistic decreases to 1, it implies the Markov chain converges to its stationary distribution. In figure 7, we report the Gelman-Rubin statistic from running 100 Markov chains on Bayesian Logistic Regression model with $n = 50$ and $m = 100$. The statistic converges to 1 quite fast, indicating a fast mixing of SGDLD. Also, we can see that using Polyak step size 0.6 mixes slightly faster than using Polyak step size 0.3 as expected.

## C.3 STOCHASTIC FACILITY LOCATION

Denote $I, J$ as the index set for facilities and customers, respectively. Denote $y_i \in \{0, 1\}$ as whether facility $i$ is open or not, $x_{ij} \in \{0, 1\}$ as whether customer $j$ is served by facility $i$, $s_i$ as the outsource for facility $i$, the objective function is

$$\min f(x, y; d) = \sum_{i \in I} c_i y_i + \sum_{i \in I, j \in J} c_{ij} d_j x_{ij} + \sum_{i \in I} g_i s_i \tag{32}$$

$$\text{s.t.} \qquad \sum_{i \in I} x_{ij} = 1, \qquad\qquad j \in J \tag{33}$$

$$\sum_{j \in J} d_{ij} x_{ij} \le K_i y_i + s_i, \qquad i \in I \tag{34}$$

$$s_i \ge 0, \qquad\qquad i \in I \tag{35}$$

where $c_i$ is fixed cost to open facility $i$, $c_{ij}$ is the transition cost from facility $i$ to customer $j$, $d_j$ is the stochastic demand of customer $j$, and $g_i$ is the outsource penalty coefficient for facility $i$, $K_i$ is the capacity for facility $i$. We set the values of these parameters following Albareda-Sambola et al. (2011); Bieniek (2015).

We run SGDLD 10k steps with batch size 1 with Polyak step size decreasing from 3.0 to 0.3. We use the temperature $\tau = 0.999^t * \tau_0$, where $\tau_0 = 200, 500, 1000$ for problems with size $15 \times 30, 40 \times 120, 100 \times 400$, respectively. After sampling, we collect the last state in each chain $(x^1, y^1), ..., (x^{100}, y^{100})$. Then we sample 10k demands $d$, and pick the $(x^*, y^*)$ that has the smallest average cost on these 10k demands $d$.

We compare SGDLD with baselines including Gurobi 10.0 (Bixby, 2007) and stochastic local search (SLS) (Hoos & Stützle, 2004) on problems in three different sizes, each has 5 instances. For Gurobi,

we use SAA with 8 samples and 1024 samples. We use the default setting of Gurobi and set a time limit for 3600 seconds. For SLS, we use the same procedure as SGDLD except for replacing the sampling step by greedily picking the best local edit in the neighborhood.

In evaluation, for each instance, we sample another 10k demands $d$ and compute the cost for the solutions obtained from different methods. We report the average cost with standard deviation on 5 instances for each method in Table 1.

### C.4 APPROXIMATE COMPUTING

We conduct the approximate computing experiments following Wang et al. (2022). One fundamental problem in approximate computing (AxC) is to assign imprecise functional units (AxC units) to execute operations such as multiplication and addition. In our experiments, we consider a computational graph that has 15 nodes of multiplication or addition that maps $\mathbb{R}^{16}$ to $\mathbb{R}$. We consider a fixed number of nodes (3, 5, or 8) are assigned to AxC units, where each unit randomly produces a result with 10% relative error in average. Denote $x \in \{0, 1\}^{15}$ as whether to assign AxC unit on node $i$. The objective is to assign the fixed number of AxC units while minimizing the expected relative error of the output $\mathbb{E}_w[f(x; w)]$, where $w$ represent a node is multiplication or addition and the randomness for all AxC units.

For each configuration $w$, we run 1 chain with 200 steps and Polyak step size decreasing from 2.0 to 0.1 to solve the problem. We use the temperature $\tau = 5 \cdot 0.96^t$ and use the final state in the chain as our solution.

### C.5 PROMPT TUNING

The prompt tuning experiment follows Wen et al. (2023). In particular, the diffusion generative models Rombach et al. (2022) can consume a text prompt to generate high-quality images and CLIP (Radford et al., 2021) can measure the similarity between the images and texts. Given a set of target images $\mathcal{I} = \{I_1, ..., I_n\}$, we sample text prompts $x$ that obtains a high CLIP score on the target images. We consider the target distribution

$$\pi(x) = \exp(\beta \mathbb{E}_{I \sim \mathcal{I}}[\text{CLIP}(x, I)]. \tag{36}$$

Following Wen et al. (2023), we run SGDLD for $T = 3000$ steps, with Polyak step size $h_t = 1$, inverse temperature $\beta_t = 0.1t$, $t = 1, ..., 3000$. In the original work, CR Wei et al. (2022) randomly initialize the text prompt and get its embedding in continuous space. Then, it applies gradient descent on the continuous embedding for 3000 steps. In the end, it converts the continuous embedding back to text prompt that is closest to the embedding. Our SGDLD directly updates the prompt $x$ to simulate the gradient flow and obtains a better result. We display more results in Figure 8.

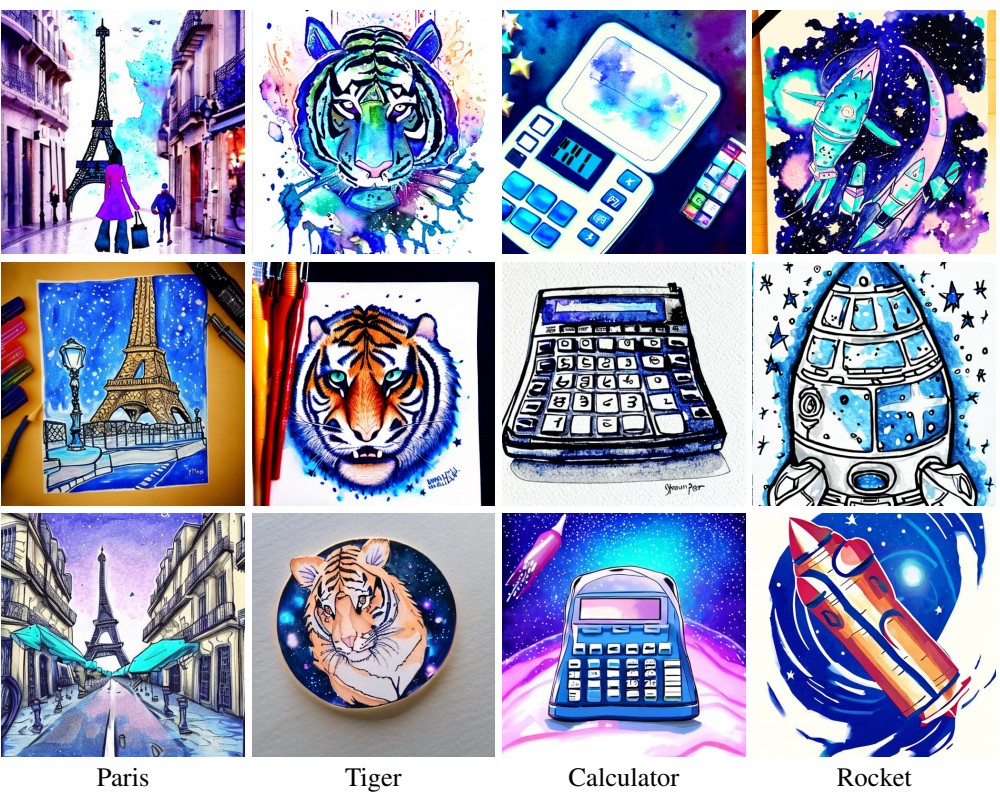

Paris      Tiger      Calculator      Rocket

Figure 8: Examples of images generated via different methods with prompt length 32: (top): SGDLD, (middle): CR (Wen et al., 2023), (bottom): SGDLD-noC

