# OpenReview forum: "Stochastic Gradient Discrete Langevin Dynamics"
_ICLR.cc/2024/Conference — Submitted to ICLR 2024_

### Official Review · Reviewer_m4Ve · 2023-10-28

**Soundness:** 2 fair
**Presentation:** 1 poor
**Contribution:** 2 fair
**Rating:** 3
**Confidence:** 2

**Summary:**

The paper under review aim to sample a target distriubtion $\pi$ over a discrete sapce $\mathcal{X}$. To this end, they consider a discrete Langevin dynamics approach from (Sun et al 2023) and (Zanella 2020). This approach consists in sampling a continuous time Markov chain with transition matrix:
$$
R(x,y) = g(\pi(y)/\pi(x)) 1_{N(x)}(y) ,
$$
where $g$ is weight function satisfying $g(a) = ag(1/a)$ and $N(x)$ is a set of neighbors of $x$.

The main challenge addressed in the paper is that in many practical scenarios, computing $g\left(\frac{\pi(y)}{\pi(x)}\right)$ can be either computationally expensive or intractable. To overcome this issue, the paper proposes a method that adapts the stochastic version of Langevin dynamics (Teh and Welling, 2011) to the discrete setting under consideration.

To achieve this adaptation, they develop a methodology for using biased estimators of the ratio $\pi(y)/\pi(x)$ through cache strategies and a discrete adaptation of the Polyak step size from optimization.

**Strengths:**

- The paper proposes interesting alternatives to the discrete Langevin dynamics from (Sun et al 2023) to address the problem of intractable target density ratio.
- The experiments demonstrate that the methodology is highly efficient and outperforms its competitors.

**Weaknesses:**

- The writing is quite poor; although the methodology is intuitive, I am still unclear about the sampling procedure implemented by the authors (please refer to my comments).
- It would have been interesting to see the proposed methodology compared to other existing approaches on simpler examples. Currently, I have some reservations about the experiments, as the results seem almost too good to be true. Toy examples would be valuable in understanding the limitations and biases introduced by the methods compared to exact MCMC algorithms, providing insight into the various approximations made by the authors in their methodology.

**Questions:**

- I suggest that the authors provide more details on what they mean by sampling from $I+\epsilon R$. It's not clear to me what the precise procedure they use.
- Similarly, I didn't understand how the Polyak step size procedure is implemented and why Equation (15) is valid.
- If I'm not mistaken, even if you sample exactly from DLD, the continuous Markov process doesn't exactly have the target distribution as the invariant distribution. Am I right? If so, this point should be highlighted.
- The statement of Proposition 4.1 lacks precision. I didn't understand what the authors mean by "when the step size ϵ decreases to 0, the sampling process associated with the jump rate from Equation 12 is asymptotically unbiased."

In conclusion, while I believe the paper presents very interesting ideas, it's not ready for publication in its current state. The presentation appears more like a draft than a properly prepared submission.

---

> ### Author Response · Authors · 2023-11-18
> **Reply to review m4Ve**
>
> We thank the reviewer for their questions. We have provided additional experiments per the request, and we kindly ask the reviewer to check our response below and see if you still have concerns.
>
> **more details on what they mean by sampling from $I + \epsilon R$**
>
> We follow the notation in [1]. Here, $R \in \mathbb{R}^{|\mathcal{X}| \times \mathcal{X}|}$ denotes the rate matrix in a continuous-time Markov chain (ctMc). Specifically, Euler’s forward method is used to simulate the ctMc. Given a simulation time denoted as $\epsilon$, $I + \epsilon R$ is used as the transition matrix of a discrete-time Markov chain. Given the current state $x \in \mathcal{X}$, the proposal distribution is given by the $x$-th row of  $I + \epsilon R$. Then we apply the standard MH to accept or reject the new state.
>
> **Toy examples would be valuable…the results seem almost too good to be true**
>
> We follow the reviewer’s suggestion to provide a toy example for better understanding of the effectiveness of the proposed method.
>
> *A new toy model*:
> We consider an $n$ dimensional binary state space $\mathcal{X} = \{0, 1\}^n$, $\pi(x) = \mathbb{E}[\pi(x|u)]$, where $\pi(x|u)$ has energy function $f(x; u) = \sum_{i=1}^n u_i x_i$ and $u_i \sim \text{Uniform}(\{-1, 1\}$ independently. This is a simplified version of the Gaussian Bernoulli Model in section 6.1. We consider $n=100$, and batch size $10$.
>
> *New experiments*:
> We compare the performance of SGDLD with standard pseudo-marginal MCMC (pm-MCMC) w.r.t. the $\ell_1$ error between the true distribution and the estimated distribution. For SGDLD, we decay the Polyak step size from $5$ to $0.5$ linearly in 50K steps. For pm-MCMC, we simply use Gibbs-1, as mentioned in the main text that the variance reduction techniques developed for continuous space do not apply in discrete space. We report the error at several different M-H steps.
>
> *Results*:
> One can see SGDLD significantly outperforms w.r.t. M-H steps. Regarding running time, pm-MCMC takes half the time of SGDLD. Hence, SGDLD beats pm-MCMC, too. Our algorithm obtains good results as it exploits the powerful gradient descent method in discrete space, whose advantages have been witnessed in many previous works [1, 2, 3, 4].
>
> | Steps | 10k | 20K | 30K | 40K | 50K |
> |-|-|-|-|-|-|
> | SGDLD | 5.49 | 3.92 | 3.17 | 2.76 | 2.43 |
> | pm-MCMC | 17.24 | 12.34 | 10.04 | 8.99 | 7.95 |
>
> **how the Polyak step size procedure is implemented and why Equation (15) is valid**
> The discrete Langevin dynamics in equation (5) gives a continuous-time Markov chain whose stationary distribution is our target distribution. One can obtain samples from the target distribution by sampling from (5) with fixed time intervals, or using adaptive time intervals with some reweightings of the obtained samples. The Polyak step size provides numerical-friendly time intervals for simulation. The reweighting technique is also used in sampling from continuous-time Markov chains [5] and SGLD [6].
>
> **Is sampling exactly from DLD unbiased**
>
> Yes, the stationary distribution of DLD is exactly the target distribution. One can show it by checking the detailed balanced condition. In particular, for a continuous-time Markov chain, it is $\pi(x) R(x, y) = \pi(y) R(y, x)$. The property of $g(\cdot)$ in equation (5) guarantees this is true.
>
> **The statement of Proposition 4.1 lacks precision**
>
> We apologize for the potential confusion here. Proposition 4.1 refers to the convergence in distribution. In particular, denoting  $\pi_\epsilon(\cdot)$ as the stationary distribution of the sampling process associated with step size $\epsilon$, then $\lim_{\epsilon \rightarrow 0} \pi_\epsilon(x) - \pi(x) = 0, \forall x \in \mathcal{X}$.
>
> > References \
> [1] Grathwohl, Will, et al. "Oops i took a gradient: Scalable sampling for discrete distributions." International Conference on Machine Learning. PMLR, 2021.\
> [2] Sun, Haoran, et al. "Path auxiliary proposal for mcmc in discrete space." International Conference on Learning Representations. 2021.\
> [3] Zhang, Ruqi, Xingchao Liu, and Qiang Liu. "A Langevin-like sampler for discrete distributions." International Conference on Machine Learning. PMLR, 2022.\
> [4] Sun, Haoran, et al. "Discrete Langevin Samplers via Wasserstein Gradient Flow." International Conference on Artificial Intelligence and Statistics. PMLR, 2023.\
> [5] Rao, Vinayak AP. Markov chain Monte Carlo for continuous-time discrete-state systems. Diss. UCL (University College London), 2012.\
> [6]] Welling, Max, and Yee W. Teh. "Bayesian learning via stochastic gradient Langevin dynamics." Proceedings of the 28th international conference on machine learning (ICML-11). 2011.

---

### Official Review · Reviewer_Ap8o · 2023-10-31

**Soundness:** 2 fair
**Presentation:** 3 good
**Contribution:** 2 fair
**Rating:** 5
**Confidence:** 4

**Summary:**

In the paper author propose a way to sample from invariant measure in discrete state space using generaliztion of SGLD to discrete spaces. Authors conduct thorough evaluation and explain how they practically made the approach work with using adaptive step-sizes, clipping and stochastic estimates. Authors also showed asymptocic results on convergence of the approach.

**Strengths:**

Quite an interesting approach to Monte-Carlo and a way to sample from the invariant measures. Well-detailed experimentations. And good emphasis on biasedness due to Jensen inequality when batches are sampled.

**Weaknesses:**

1. Main claims of the paper are asympototic, it is generally hard to follow and check whether they are correct.
2. References to many claims are missing.
3. Some minor typos in equations (see questions)

**Questions:**

1. Equation 2. The Wiener process in discrete steps is indexed with timestep and gradient looking into the future, summation over N and this is equation for $x_{t+h}$ depending on $x_{t+nh}$. I honestly can't make sense of it. I assume the authors made some honest typos there as were in a hurry with submission. Can you please explain/fix it as this is standard SGLD (and I assume that this is what authors wanted to write there).

2. Likewise entire section 2 contains a lot of statements with under "mild conditions", "easy to show", "asymptotically" (in what sense). I'd like to see some references and examples what is meant by easy, mild and asymptotic, as for applications that concern Bayesian inference one might be interested in having guarantee of convergence up to p-th moment (which is also my question to the proposed method -- convergence that is asymptotic -- it is just in probability?), while what is easy to show under mild conditions is convergence in probability that is unaplicable to practical setups. Basically, some references are needed here.

3. Appendix Equation 27. Should not there be minus sign before xWx as otherwise density is non-normalizable and henceforth this is not valid distribution?

4. Propositon A.1 seems to be unapplicable to example in Section C.2. Generally, limitations of assumptions and their applicability to the examples is not shown.

5. Out of curiosity, why this is called as SGDLD? This looks more like broader MCMC, whilst Langevin Dynamics is about stochastic differential equations (that are driven by some continious noise), while here it is justified by just showing that under certain conditions Kolmogorov equation gives invariant measure, however, this does look to me just some other form of MCMC rather than LD, as Kolmogorov equations are not about just LD. (nevertheless, this is interesting form of MCMC)

---

> ### Author Response · Authors · 2023-11-18
> **Reply to reviewer Ap8o**
>
> We thank the reviewer for their insightful questions. Please see our response below and we look forward to learning your thoughts on these.
>
> **Equation 2. The Wiener process in discrete steps is indexed with timestep and gradient looking into the future**
>
> We use equation 2 to denote a telescoping sum: $x_{t+(i+1)\epsilon} = x_{t+i\epsilon} + \epsilon \nabla \hat{f}(x_{t+ i\epsilon}) + \sqrt{2}W_\epsilon$ for i = $0, …, N-1$. When $h$ is sufficiently small, one can reduce the telescoping sum via equation 3 to prove that SGLD is asymptotically unbiased. In the SGLD paper [1], their equation 8 also made a similar statement.
>
> **Mild Condition and Asymptotically in section 2 / Exampling in section 2**
>
> Yes, proposition 4.1 only shows the convergence in distribution. This work aims for an efficient sampling framework in discrete space by combining the SGLD and the recent advances in discrete sampling. Current proof relies on the central limit theorem which converges at the rate of $N^{-\frac{1}{2}}$. We believe the theoretical part of Proposition 1 can be improved in the future work, and we will discuss the limitations of the current version in revision.
>
> **Appendix Equation 27**
>
> Yes, a minus in front of $xWx$ is missing. Thank you for catching the typo!
>
> **why this is called as SGDLD**
>
> You are correct, this is a broader case of MCMC. The name is after several recent discrete sampling works [2, 3, 4], where the designed samplers in discrete space can be viewed as a discretization of a Wasserstein gradient flow. Since this is similar to LD, [3] named it discrete Langevin dynamics and we follow the same convention.
>
> > References \
> [1] Welling, Max, and Yee W. Teh. "Bayesian learning via stochastic gradient Langevin dynamics." Proceedings of the 28th international conference on machine learning (ICML-11). 2011.\
> [2] Zanella, Giacomo. "Informed proposals for local MCMC in discrete spaces." Journal of the American Statistical Association (2019).\
> [3] Grathwohl, Will, et al. "Oops i took a gradient: Scalable sampling for discrete distributions." International Conference on Machine Learning. PMLR, 2021.\
> [4] Sun, Haoran, et al. "Discrete Langevin Samplers via Wasserstein Gradient Flow." International Conference on Artificial Intelligence and Statistics. PMLR, 2023.

---

### Official Review · Reviewer_i63H · 2023-11-08

**Soundness:** 3 good
**Presentation:** 3 good
**Contribution:** 2 fair
**Rating:** 6
**Confidence:** 2

**Summary:**

This paper proposes a sampling strategy, Stochastic Gradient Discrete Langevin Dynamics, for a more efficient and accurate MCMC sampling in discrete spaces. This strategy contains a cache method and a modified Polyak step size.

**Strengths:**

1. The authors identify the problem when sampling in a discrete space.

2. To decrease the bias, they propose a caching technique that expands the batch size with no extra computation cost.

3. To make the algorithm more stable, an adaptive step size method is introduced.

4. Many experiments are done to verify their claims.

**Weaknesses:**

1. Will the caching technique require a lot of memory? If yes, is there a way to make the cache more memory-efficient?

2. Could you be more precise about how $N_2$ controls the MC error in equations (9) and (10)?

**Questions:**

see weaknesses.

---

> ### Author Response · Authors · 2023-11-18
> **Reply to reviewer i63H regarding the two questions**
>
> Thank you so much for your review and your constructive questions. Please see our response below:
>
> **Will the caching technique require a lot of memory? If yes, is there a way to make the cache more memory-efficient?**
>
> In caching, we only need to store $\sum_{i=1}^N \pi(y|u_i), \forall y \in \mathcal{N}(x)$ for quenched model and $\sum_{i=1}^N f(y|u_i), \forall y \in \mathcal{N}(x)$ for Bayesian model, where $\pi(\cdot|u)$ is the unnormalized probability, $f(\cdot|u)$ is the energy function, and $\mathcal{N}(x)$ is the neighborhood of the current state $x$. Previous works [1, 2, 3, 4] show that one only needs to consider a 1-Hamming ball Neighborhood to simulate the Langevin Dynamics. Hence, for a model with $n$ variables and $C$ categories, one only needs to store $O(nC)$ scalars in memory, which is not a high cost for memory.
>
> **Could you be more precise about how $N_2$ controls the MC error in equations (9) and (10)?**
>
> In section 3.2, we assume we have an unbiased estimator of the rate matrix $\mathbb{R}^{|\mathcal{X}| \times |\mathcal{X}|}$. That is to say, one has $\mathbb{E_u}[\hat{R_{i, j}}(u)] = R_{i, j}$, where $R$ denotes the ground truth rate matrix, $\hat{R}(u)$ is a finite sample estimate based on $u$, and $i, j \in \mathcal{X}$ are two arbitrary state. Since we also assume $\|\hat{R} - R\|$ is bounded, by central limit theory, with high probability, $|\frac{1}{N} \sum_{i=1}^N \hat{R_{i, j}}(u_i) - R_{i, j}| = O(\frac{1}{\sqrt{N}})$. Since we consider a finite space $\mathcal{X}$, we have $\frac{1}{N_2} \sum_{j=1}^{N_2} R_{iN_1 + j} = R + O(\frac{1}{\sqrt{N_2}})$.
>
> > References \
> [1] Grathwohl, Will, et al. "Oops i took a gradient: Scalable sampling for discrete distributions." International Conference on Machine Learning. PMLR, 2021.\
> [2] Sun, Haoran, et al. "Path auxiliary proposal for mcmc in discrete space." International Conference on Learning Representations. 2021.\
> [3] Zhang, Ruqi, Xingchao Liu, and Qiang Liu. "A Langevin-like sampler for discrete distributions." International Conference on Machine Learning. PMLR, 2022.\
> [4] Sun, Haoran, et al. "Discrete Langevin Samplers via Wasserstein Gradient Flow." International Conference on Artificial Intelligence and Statistics. PMLR, 2023.

---

> > ### Comment · Reviewer_i63H · 2023-11-22
> >
> > I would like to thank the authors for addressing my concerns and questions. I am keeping my rating unchanged.

---

### Meta-Review · Area_Chair_bfEa · 2023-12-14

**Metareview:**

This paper proposes a new discrete version of stochastic gradient Langevin dynamics. More concretely, they propose to use of a stochastic gradient cache and Polyak step size. The method is verified by numerical experiments.

Although this paper proposes a new approach, the writing of the paper is not ready for publication. In particular, several mathematical statements are not presented in a well defined manner and those are presented without rigorous justification. This paper requires a substantial revision before publication. Hence, I cannot recommend acceptance for this paper.

**Justification For Why Not Higher Score:**

The writing of this paper requires substantial revision before publication. Although the proposal itself would be interesting, it is hard to verify its justification.

**Justification For Why Not Lower Score:**

N/A

---

### Decision · Program_Chairs · 2024-01-16

Reject